# Violaceous Nodules on the Left Forearm of an Immunosuppressed Patient Following Heart Transplantation for Cardiac Amyloidosis

**DOI:** 10.3390/dermatopathology12010002

**Published:** 2025-01-16

**Authors:** Zachary Corey, Lydia A. Luu, Sabrina Newman, Shyam S. Raghavan

**Affiliations:** 1Department of Pathology, University of Colorado School of Medicine, Aurora, CO 80045, USA; shyam.raghavan@cuanschutz.edu; 2Department of Dermatology, University of Colorado School of Medicine, Aurora, CO 80045, USA; lydia.luu@cuanschutz.edu (L.A.L.); sabrina.newman@cuanschutz.edu (S.N.)

**Keywords:** mucormycosis, *Mucorales*, *Rhizopus*, dermatomycosis, zygomycosis

## Abstract

We present the case of a 60-year-old immunocompromised man who presented with two pruritic pink–red indurated nodules with overlying scale and focal areas of ulceration on his left dorsal and left medial forearm, which evolved over a 2-month period. The pathology showed numerous fungal hyphae present that were pauci-septate with various branched angles and variable hyphal thickness. Fungal cultures grew *Rhizopus* species and a universal fungal PCR detected the *Rhizopus oryzae* complex. Based on the clinicopathologic correlation, the diagnosis of cutaneous mucormycosis was made. Cutaneous mucormycosis is an aggressive fungal infection of the *Mucorales* family occurring after the inoculation of fungal spores in disrupted skin. It usually presents as a necrotic eschar but can also present as cellulitis that evolves into a necrotic ulcer. A prompt diagnosis is critical for the effective management of cutaneous mucormycosis. The treatment includes an immediate systemic treatment with amphotericin B and a surgical debridement of the necrotic regions. Given the wide range of presenting symptoms, clinical suspicion for this emergent condition must remain high in immunocompromised and diabetic patients.

## 1. Case Presentation

A 60-year-old man who was on tacrolimus and mycophenolate after a heart transplant for amyloidosis two months prior presented with two nodules on his left dorsal and left medial forearm. The nodules appeared following a dog scratch and evolved over a few weeks to two nodules with focal ulceration (Figure 1). Two courses of antibiotics, amoxicillin and trimethoprim-sulfamethoxazole, failed to resolve the symptoms.

A physical exam revealed two pink–red indurated nodules with overlying scale and focal areas of ulceration on the left dorsal and left medial forearm. The patient denied pain on palpation but endorsed significant pruritus. Punch biopsies were performed on the left forearm nodules (Figure 2). A further diagnostic workup included fungal culture, fungal PCR testing and fungal serology.

Of note, the patient’s postoperative course from his heart transplant was complicated by steroid-induced hyperglycemia. Three days following his discharge, the patient presented to the ED with a blood glucose of 351 due to a difficulty with insulin adherence, which was subsequently corrected.

## 2. What Is the Diagnosis?

Cutaneous Squamous Cell CarcinomaPrimary Cutaneous AmyloidosisCutaneous MucormycosisAspergillusBlastomycosis

## 3. Diagnosis

C.Cutaneous Mucormycosis

## 4. Discussion

Histopathology sections show skin with an acanthotic and spongiotic epidermis. Within the dermis, there is mixed inflammation along with granulomatous foci and giant cells. Numerous fungal hyphae that are pauci-septate are present, with various branched angles and variable hyphal thickness (Figure 2). These were highlighted on GMS stains. Fungal cultures grew *Rhizopus* species and a universal fungal PCR detected the *Rhizopus oryzae* complex. The remainder of the infectious workup results indicated negative (1,3)-Beta-D-Glucan, *Aspergillus*/other Galactomannan Ag, and *Coccidioides* IgG or IgA antibody by immunodiffusion.

Mucormycosis is an aggressive opportunistic fungal infection of the *Mucorales* order (e.g., *Rhizopus* spp., *Mucor* spp., and *Lichtheimia* spp.) [1,2,3]. Cutaneous mucormycosis is the third most common type, following the pulmonary and rhinocerebral types Infection occurs following the direct inoculation of fungal spores into disrupted skin or via hematogenous spread [1,3,4]. Immunocompromised patients, and those in a state of increased iron (which can occur in diabetes mellitus, following uncontrolled hyperglycemia) are particularly susceptible to infection [4,5,6]. Clinically, cutaneous mucormycosis usually presents as a necrotic eschar. Less commonly, it can also present as cellulitis that evolves into a necrotic ulcer [5,6]. Early diagnosis is critical; however, it can be delayed by a nonspecific and uncommon presentation, as seen in this case [1].

This patient had a history of solid-organ transplant complicated by steroid-induced hyperglycemia and presented with two pruritic pink–red nodules on his left forearm following a dog scratch. Squamous cell carcinoma (SCC) can present as firm, scaly, pink–red nodules following trauma, but histologically, it would have squamous epithelial cells in the epidermis extending into the dermis [7]. Based on the patient’s history of amyloidosis, cutaneous amyloidosis is a possibility, but that would also be expected to have a scattered array of pink amorphous globular deposits within the papillary dermis [8]. In this case, the broad hyphae and irregular branching pattern, coupled with the PCR and fungal cultures, support the diagnosis of cutaneous mucormycosis. Aspergillus, which can present similarly, and other septate hyaline molds would have a more regular branching pattern and thinner septa compared to mucormycosis [1,3,6]. While Blastomycosis can also be found in immunocompromised hosts, with a clinical presentation that varies but can include papulopustular lesions that progress to verrucous lesions, it exhibits broad-based budding with thick, double-refractile walls [9]. Table 1 presents a comparison of these fungal infections. We suggest that this case represents a unique presentation of cutaneous mucormycosis.

Prompt diagnosis is critical for the effective management of cutaneous mucormycosis. Given the wide range of presenting symptoms, clinical suspicion for this entity must remain high in immunocompromised and diabetic patients. Suspected and confirmed mucormycosis is a medical emergency. Lipid amphotericin B formulation is the systemic therapy of choice and should be initiated immediately following the suspicion of mucormycosis [1,4]. The surgical debridement of necrotic regions is also recommended to improve patient outcomes and prevent dissemination. Cutaneous mucormycosis should be diagnosed via histopathology to identify the characteristic hyphae of *Mucorales*. Tissue culture and molecular techniques may be used to identify the genus and species and confirm the diagnosis in inconspicuous cases [4]. In cases where possible, efforts should also be spent on reversing the underlying immunosuppression or correction of diabetic ketoacidosis [4]. In this case, amphotericin B and surgical debridement were initiated promptly following diagnosis. The pathology revealed clear margins, and the patient is scheduled to undergo a skin graft repair with plastic surgery.

## Figures and Tables

**Figure 1 dermatopathology-12-00002-f001:**
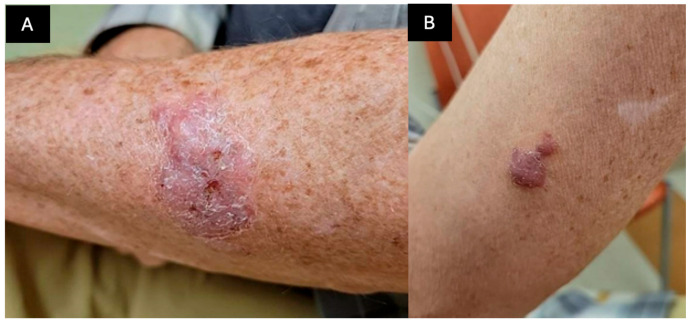
Clinical presentation: two pink-red indurated nodules with overlying scale and focal areas of ulceration on the (**A**) left dorsal forearm and (**B**) left ventral forearm.

**Figure 2 dermatopathology-12-00002-f002:**
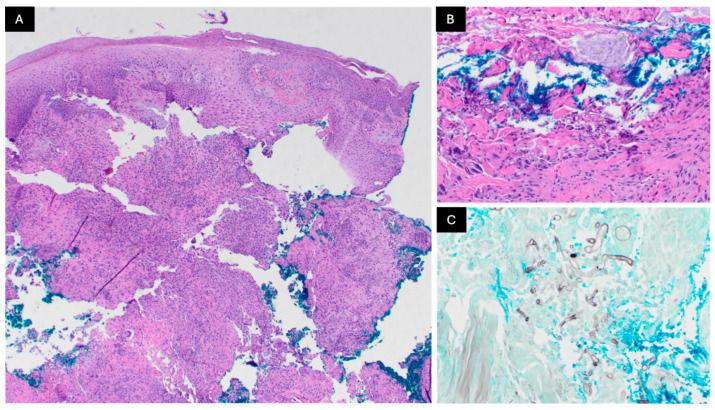
Cutaneous mucormycosis histopathology: (**A**) a 4× H&E of biopsy specimen revealing mixed inflammation along with granulomatous foci and giant cells within the dermis, (**B**) a 40× H&E of biopsy showcasing fungal hyphae, and (**C**) a 20× GMS stain of biopsy specimen highlighting the fungal hyphae, which are pauci-septate with various branched angles and variable hyphal thickness.

**Table 1 dermatopathology-12-00002-t001:** Differential diagnosis of cutaneous fungal infections.

Cutaneous Fungal Infection	Clinical Presentation	Histomorphology
Mucormycosis	Immunocompromised patient with a necrotic eschar or, less commonly, cellulitis that evolves into a necrotic ulcer.	Broad, non-septate hyphae with an irregular branching pattern and an angle of branching close to 90°
Aspergillus	Immunocompromised patient with erythematous-violaceous plaque or nodule at the site of skin trauma (e.g., venipuncture site) with a central necrotic eschar.	Thin, septate hyphae with a regular branching pattern and angle of branching close to 45°
Blastomycosis	Immunocompromised patient with subcutaneous nodules or papules that progress over weeks to months into ulcers and crusted sores.	Round, yeast cells with broad based budding with a thick, double-refractile wall

## Data Availability

No new data were created or analyzed in this study. Data sharing is not applicable to this article.

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
