# Peer review of "Violaceous Nodules on the Left Forearm of an Immunosuppressed Patient Following Heart Transplantation for Cardiac Amyloidosis"

_dermatopathology, 2025, doi:10.3390/dermatopathology12010002_

Round 1
Reviewer 1 Report
Comments and Suggestions for Authors
"it can also present as cellulitis that evolves into an ecthyma-like nodule ..." I do not really understand: ecthyma is characterized by an ulcer, so how can cellulitis which is defined as a diffuse inflammation, evolve into a nodule that is an ulcer?
"Early diagnosis is critical; however, it can be delayed by a nonspecific and uncommon presentation, as seen in this case" It depends on the duration of infection how the clinical aspect looks like, and this actually the reason to take the biopsy. In my experience this is not an "uncommon presentation", particularly not concerning the histopathology.
"open heart transplantation" Can it be done "closed"?
Author Response
Comment 1: "it can also present as cellulitis that evolves into an ecthyma-like nodule ..." I do not really understand: ecthyma is characterized by an ulcer, so how can cellulitis which is defined as a diffuse inflammation, evolve into a nodule that is an ulcer?
Response 1: Thank you for bringing up this potential area of confusion. We have altered the text to be more clear by re-writing this statement to “it can also present as cellulitis that evolves into a necrotic” (lines18; 77-78).
Comment 2: "Early diagnosis is critical; however, it can be delayed by a nonspecific and uncommon presentation, as seen in this case" It depends on the duration of infection how the clinical aspect looks like, and this actually the reason to take the biopsy. In my experience this is not an "uncommon presentation", particularly not concerning the histopathology.
Response 2: Thank you for raising this concern. In our experience, mucormycosis more commonly presents clinically as an angioinvasive fungal infection of the sinus tract. We recognize that personal experience may vary, and have decided to keep the sentence as is. We appreciate this thoughtful feedback.
Comment 3: "open heart transplantation" Can it be done "closed"?
Response 3: We have deleted “open” in the title (Line 3), line 27, and line 39.
Reviewer 2 Report
Comments and Suggestions for Authors
In the differential diagnosis why not mention maduromycosis instead of SCC clinically different and with a chronic slow growing. It is possible an early detection by KOH? Because this is a non specific and an uncommon presentation let we know the other signs of presentation useful for a critical early diagnosis, and the possible other localization. Only amphotericin B is used or do you have experience with second generation azoles? Do you know mortality rate?
Author Response
Comment 1: In the differential diagnosis why not mention maduromycosis instead of SCC clinically different and with a chronic slow growing. It is possible an early detection by KOH? Because this is a non specific and an uncommon presentation let we know the other signs of presentation useful for a critical early diagnosis, and the possible other localization. Only amphotericin B is used or do you have experience with second-generation azoles? Do you know mortality rate?
Response 1: We greatly value this suggestion. However, our clinical team specifically noted that they were concerned for SCC in this case, so we have opted to keep SCC as part of the potential diagnosis and discussion. There are reports of using KOH to detect cutaneous mucormycosis (PMID: 34897955); however, evaluation by histopathology is the gold standard method for diagnosis. We discuss the potential presenting signs for cutaneous mucormycosis in lines 76-78, and mention that mucormycosis otherwise typically localizes to the pulmonary system or sinonasal tract in line 72. Amphotericin B is the treatment of choice for mucormycosis due to the aggressive nature of the infection. There is some evidence for the use of isavuconazole in the treatment of mucormycosis (PMID: 29750016) but it is not the recommendation we would make to readers at this time.
Reviewer 3 Report
Comments and Suggestions for Authors
I have reviewed the manuscript: "Clinicopathologic Challenge: Violaceous Nodules on the Left Forearm of an Immunosuppressed Patient Following Open Heart Transplantation for Cardiac Amyloidosis." I have the following suggestions:
1. Line 44 reads: "All figures and tables should be cited in the main text as Figure 1, Table 1, etc." Please correct/delete that.
2. Fig 2C. GMS stain. I can see the fungi, however, the image is off center and low power. Please provide a better image so that the reader can evaluate the morphology of the fungus.
3. A detailed table, which compares the clinical presentation and histomorphological appearance of fungal infections included in the differential diagnosis, can be informative for the readers.
Thank you.
Author Response
I have reviewed the manuscript: "Clinicopathologic Challenge: Violaceous Nodules on the Left Forearm of an Immunosuppressed Patient Following Open Heart Transplantation for Cardiac Amyloidosis." I have the following suggestions:
Comment 1: Line 44 reads: "All figures and tables should be cited in the main text as Figure 1, Table 1, etc." Please correct/delete that.
Response 1: Thank you for noting this. This has been deleted from the text.
Comment 2: Fig 2C. GMS stain. I can see the fungi, however, the image is off center and low power. Please provide a better image so that the reader can evaluate the morphology of the fungus.
Response 2: Thank you for that suggestion. The figure has been amended in the text to allow for a better evaluation of the fungal morphology.
Comment 3: A detailed table, which compares the clinical presentation and histomorphological appearance of fungal infections included in the differential diagnosis, can be informative for the readers.
Response3 : We greatly appreciate this suggestion. We have updated the text to include this table. Thank you!
Thank you.